# Training Quantized Nets: A Deeper Understanding

**Hao Li**[1]*, **Soham De**[1]*, **Zheng Xu**[1], **Christoph Studer**[2], **Hanan Samet**[1], **Tom Goldstein**[1]
[1]Department of Computer Science, University of Maryland, College Park
[2]School of Electrical and Computer Engineering, Cornell University
{haoli,sohamde,xuzh,hjs,tomg}@cs.umd.edu, studer@cornell.edu

## Abstract

Currently, deep neural networks are deployed on low-power portable devices by first training a full-precision model using powerful hardware, and then deriving a corresponding low-precision model for efficient inference on such systems. However, training models directly with coarsely quantized weights is a key step towards learning on embedded platforms that have limited computing resources, memory capacity, and power consumption. Numerous recent publications have studied methods for training quantized networks, but these studies have mostly been empirical. In this work, we investigate training methods for quantized neural networks from a theoretical viewpoint. We first explore accuracy guarantees for training methods under convexity assumptions. We then look at the behavior of these algorithms for non-convex problems, and show that training algorithms that exploit high-precision representations have an important greedy search phase that purely quantized training methods lack, which explains the difficulty of training using low-precision arithmetic.

## 1   Introduction

Deep neural networks are an integral part of state-of-the-art computer vision and natural language processing systems. Because of their high memory requirements and computational complexity, networks are usually trained using powerful hardware. There is an increasing interest in training and deploying neural networks directly on battery-powered devices, such as cell phones or other platforms. Such low-power embedded systems are memory and power limited, and in some cases lack basic support for floating-point arithmetic.

To make neural nets practical on embedded systems, many researchers have focused on training nets with *coarsely quantized* weights. For example, weights may be constrained to take on integer/binary values, or may be represented using low-precision (8 bits or less) fixed-point numbers. Quantized nets offer the potential of superior memory and computation efficiency, while achieving performance that is competitive with state-of-the-art high-precision nets. Quantized weights can dramatically reduce memory size and access bandwidth, increase power efficiency, exploit hardware-friendly bitwise operations, and accelerate inference throughput [1–3].

Handling low-precision weights is difficult and motivates interest in new training methods. When learning rates are small, stochastic gradient methods make small updates to weight parameters. Binarization/discretization of weights after each training iteration "rounds off" these small updates and causes training to stagnate [1]. Thus, the naïve approach of quantizing weights using a rounding procedure yields poor results when weights are represented using a small number of bits. Other approaches include classical stochastic rounding methods [4], as well as schemes that combine full-precision floating-point weights with discrete rounding procedures [5]. While some of these schemes seem to work in practice, results in this area are largely experimental, and little work has been devoted to explaining the excellent performance of some methods, the poor performance of others, and the important differences in behavior between these methods.

**Contributions** This paper studies quantized training methods from a theoretical perspective, with the goal of understanding the differences in behavior, and reasons for success or failure, of various methods. In particular, we present a convergence analysis showing that classical stochastic rounding (SR) methods [4] as well as newer and more powerful methods like BinaryConnect (BC) [5] are capable of solving convex discrete problems up to a level of accuracy that depends on the quantization level. We then address the issue of why algorithms that maintain floating-point representations, like BC, work so well, while fully quantized training methods like SR stall before training is complete. We show that the long-term behavior of BC has an important annealing property that is needed for non-convex optimization, while classical rounding methods lack this property.

## 2 Background and Related Work

The arithmetic operations of deep networks can be truncated down to 8-bit fixed-point without significant deterioration in inference performance [4, 6–9]. The most extreme scenario of quantization is binarization, in which only 1-bit (two states) is used for weight representation [10, 5, 1, 3, 11, 12].

Previous work on obtaining a quantized neural network can be divided into two categories: quantizing pre-trained models with or without retraining [7, 13, 6, 14, 15], and training a quantized model from scratch [4, 5, 3, 1, 16]. We focus on approaches that belong to the second category, as they can be used for both training and inference under constrained resources.

For training quantized NNs from scratch, many authors suggest maintaining a high-precision floating point copy of the weights while feeding quantized weights into backprop [5, 11, 3, 16], which results in good empirical performance. There are limitations in using such methods on low-power devices, however, where floating-point arithmetic is not always available or not desirable. Another widely used solution using only low-precision weights is *stochastic rounding* [17, 4]. Experiments show that networks using 16-bit fixed-point representations with stochastic rounding can deliver results nearly identical to 32-bit floating-point computations [4], while lowering the precision down to 3-bit fixed-point often results in a significant performance degradation [18]. Bayesian learning has also been applied to train binary networks [19, 20]. A more comprehensive review can be found in [3].

## 3 Training Quantized Neural Nets

We consider empirical risk minimization problems of the form:

$$\min_{w \in \mathcal{W}} F(w) := \frac{1}{m} \sum_{i=1}^{m} f_i(w), \tag{1}$$

where the objective function decomposes into a sum over many functions $f_i : \mathbb{R}^d \to \mathbb{R}$. Neural networks have objective functions of this form where each $f_i$ is a non-convex loss function. When floating-point representations are available, the standard method for training neural networks is stochastic gradient descent (SGD), which on each iteration selects a function $\tilde{f}$ randomly from $\{f_1, f_2, \ldots, f_m\}$, and then computes

$$\text{SGD: } w^{t+1} = w^t - \alpha_t \nabla \tilde{f}(w^t), \tag{2}$$

for some learning rate $\alpha_t$. In this paper, we consider the problem of training convolutional neural networks (CNNs). Convolutions are computationally expensive; low precision weights can be used to accelerate them by replacing expensive multiplications with efficient addition and subtraction operations [3, 9] or bitwise operations [11, 16].

To train networks using a low-precision representation of the weights, a quantization function $Q(\cdot)$ is needed to convert a real-valued number $w$ into a quantized/rounded version $\hat{w} = Q(w)$. We use the same notation for quantizing vectors, where we assume $Q$ acts on each dimension of the vector. Different quantized optimization routines can be defined by selecting different quantizers, and also by selecting when quantization happens during optimization. The common options are:

**Deterministic Rounding (R)** A basic *uniform* or deterministic quantization function snaps a floating point value to the closest quantized value as:

$$Q_d(w) = \text{sign}(w) \cdot \Delta \cdot \left\lfloor \frac{|w|}{\Delta} + \frac{1}{2} \right\rfloor, \tag{3}$$

where $\Delta$ denotes the quantization step or resolution, i.e., the smallest positive number that is representable. One exception to this definition is when we consider binary weights, where all weights are constrained to have two values $w \in \{-1, 1\}$ and uniform rounding becomes $Q_d(w) = \text{sign}(w)$.

The deterministic rounding SGD maintains quantized weights with updates of the form:

$$\text{Deterministic Rounding: } w_b^{t+1} = Q_d\big(w_b^t - \alpha_t \nabla \tilde{f}(w_b^t)\big), \qquad (4)$$

where $w_b$ denotes the low-precision weights, which are quantized using $Q_d$ immediately after applying the gradient descent update. If gradient updates are significantly smaller than the quantization step, this method loses gradient information and weights may never be modified from their starting values.

**Stochastic Rounding (SR)**   The quantization function for *stochastic rounding* is defined as:

$$Q_s(w) = \Delta \cdot \begin{cases} \lfloor \frac{w}{\Delta} \rfloor + 1 & \text{for } p \leq \frac{w}{\Delta} - \lfloor \frac{w}{\Delta} \rfloor, \\ \lfloor \frac{w}{\Delta} \rfloor & \text{otherwise}, \end{cases} \qquad (5)$$

where $p \in [0, 1]$ is produced by a uniform random number generator. This operator is non-deterministic, and rounds its argument up with probability $w/\Delta - \lfloor w/\Delta \rfloor$, and down otherwise. This quantizer satisfies the important property $\mathbb{E}[Q_s(w)] = w$. Similar to the deterministic rounding method, the SR optimization method also maintains quantized weights with updates of the form:

$$\text{Stochastic Rounding: } w_b^{t+1} = Q_s\big(w_b^t - \alpha_t \nabla \tilde{f}(w_b^t)\big). \qquad (6)$$

**BinaryConnect (BC)**   The BinaryConnect algorithm [5] accumulates gradient updates using a full-precision buffer $w_r$, and quantizes weights just before gradient computations as follows.

$$\text{BinaryConnect: } w_r^{t+1} = w_r^t - \alpha_t \nabla \tilde{f}\big(Q(w_r^t)\big). \qquad (7)$$

Either stochastic rounding $Q_s$ or deterministic rounding $Q_d$ can be used for quantizing the weights $w_r$, but in practice, $Q_d$ is the common choice. The original BinaryConnect paper constrains the low-precision weights to be $\{-1, 1\}$, which can be generalized to $\{-\Delta, \Delta\}$. A more recent method, Binary-Weights-Net (BWN) [3], allows different filters to have different scales for quantization, which often results in better performance on large datasets.

**Notation**   For the rest of the paper, we use $Q$ to denote both $Q_s$ and $Q_d$ unless the situation requires this to be distinguished. We also drop the subscripts on $w_r$ and $w_b$, and simply write $w$.

## 4   Convergence Analysis

We now present convergence guarantees for the Stochastic Rounding (SR) and BinaryConnect (BC) algorithms, with updates of the form (6) and (7), respectively. For the purposes of deriving theoretical guarantees, we assume each $f_i$ in (1) is differentiable and convex, and the domain $\mathcal{W}$ is convex and has dimension $d$. We consider both the case where $F$ is $\mu$-strongly convex: $\langle \nabla F(w'), w - w' \rangle \leq F(w) - F(w') - \frac{\mu}{2}\|w - w'\|^2$, as well as where $F$ is weakly convex. We also assume the (stochastic) gradients are bounded: $\mathbb{E}\|\nabla \tilde{f}(w^t)\|^2 \leq G^2$. Some results below also assume the domain of the problem is finite. In this case, the rounding algorithm clips values that leave the domain. For example, in the binary case, rounding returns bounded values in $\{-1, 1\}$.

### 4.1   Convergence of Stochastic Rounding (SR)

We can rewrite the update rule (6) as:

$$w^{t+1} = w^t - \alpha_t \nabla \tilde{f}(w^t) + r^t,$$

where $r^t = Q_s(w^t - \alpha_t \nabla \tilde{f}(w^t)) - w^t + \alpha_t \nabla \tilde{f}(w^t)$ denotes the quantization error on the $t$-th iteration. We want to bound this error in expectation. To this end, we present the following lemma.

**Lemma 1.** *The stochastic rounding error $r^t$ on each iteration can be bounded, in expectation, as:*

$$\mathbb{E}\|r^t\|^2 \leq \sqrt{d}\Delta \alpha_t G,$$

*where $d$ denotes the dimension of $w$.*

Proofs for all theoretical results are presented in the Appendices. From Lemma 1, we see that the rounding error per step decreases as the learning rate $\alpha_t$ decreases. This is intuitive since the probability of an entry in $w^{t+1}$ differing from $w^t$ is small when the gradient update is small relative to $\Delta$. Using the above lemma, we now present convergence rate results for Stochastic Rounding (SR) in both the strongly-convex case and the non-strongly convex case. Our error estimates are ergodic, i.e., they are in terms of $\bar{w}^T = \frac{1}{T}\sum_{t=1}^T w^t$, the average of the iterates.

**Theorem 1.** *Assume that $F$ is $\mu$-strongly convex and the learning rates are given by $\alpha_t = \frac{1}{\mu(t+1)}$. Consider the SR algorithm with updates of the form* (6). *Then, we have:*

$$\mathbb{E}[F(\bar{w}^T) - F(w^\star)] \leq \frac{(1+\log(T+1))G^2}{2\mu T} + \frac{\sqrt{d}\Delta G}{2},$$

*where $w^\star = \arg\min_w F(w)$.*

**Theorem 2.** *Assume the domain has finite diameter $D$, and learning rates are given by $\alpha_t = \frac{c}{\sqrt{t}}$, for a constant $c$. Consider the SR algorithm with updates of the form* (6). *Then, we have:*

$$\mathbb{E}[F(\bar{w}^T) - F(w^\star)] \leq \frac{1}{c\sqrt{T}}D^2 + \frac{\sqrt{T+1}}{2T}cG^2 + \frac{\sqrt{d}\Delta G}{2}.$$

We see that in both cases, SR converges until it reaches an "accuracy floor." As the quantization becomes more fine grained, our theory predicts that the accuracy of SR approaches that of high-precision floating point at a rate linear in $\Delta$. This extra term caused by the discretization is unavoidable since this method maintains quantized weights.

## 4.2 Convergence of Binary Connect (BC)

When analyzing the BC algorithm, we assume that the Hessian satisfies the Lipschitz bound: $\|\nabla^2 f_i(x) - \nabla^2 f_i(y)\| \leq L_2 \|x - y\|$ for some $L_2 \geq 0$. While this is a slightly non-standard assumption, we will see that it enables us to gain better insights into the behavior of the algorithm.

The results here hold for both stochastic and uniform rounding. In this case, the quantization error $r$ does not approach 0 as in SR-SGD. Nonetheless, the effect of this rounding error diminishes with shrinking $\alpha_t$ because $\alpha_t$ multiplies the gradient update, and thus implicitly the rounding error as well.

**Theorem 3.** *Assume $F$ is $L$-Lipschitz smooth, the domain has finite diameter $D$, and learning rates are given by $\alpha_t = \frac{c}{\sqrt{t}}$. Consider the BC-SGD algorithm with updates of the form* (7). *Then, we have:*

$$\mathbb{E}[F(\bar{w}^T) - F(w^\star)] \leq \frac{1}{2c\sqrt{T}}D^2 + \frac{\sqrt{T+1}}{2T}cG^2 + \sqrt{d}\Delta LD.$$

As with SR, BC can only converge up to an error floor. So far this looks a lot like the convergence guarantees for SR. However, things change when we assume strong convexity and bounded Hessian.

**Theorem 4.** *Assume that $F$ is $\mu$-strongly convex and the learning rates are given by $\alpha_t = \frac{1}{\mu(t+1)}$. Consider the BC algorithm with updates of the form* (7). *Then we have:*

$$\mathbb{E}[F(\bar{w}^T) - F(w^\star)] \leq \frac{(1+\log(T+1))G^2}{2\mu T} + \frac{DL_2\sqrt{d}\Delta}{2}.$$

Now, the error floor is determined by both $\Delta$ and $L_2$. For a quadratic least-squares problem, the gradient of $F$ is linear and the Hessian is constant. Thus, $L_2 = 0$ and we get the following corollary.

**Corollary 1.** *Assume that $F$ is quadratic and the learning rates are given by $\alpha_t = \frac{1}{\mu(t+1)}$. The BC algorithm with updates of the form* (7) *yields*

$$\mathbb{E}[F(\bar{w}^T) - F(w^\star)] \leq \frac{(1+\log(T+1))G^2}{2\mu T}.$$

We see that the real-valued weights accumulated in BC can converge to the *true minimizer* of quadratic losses. Furthermore, this suggests that, when the function behaves like a quadratic on the distance

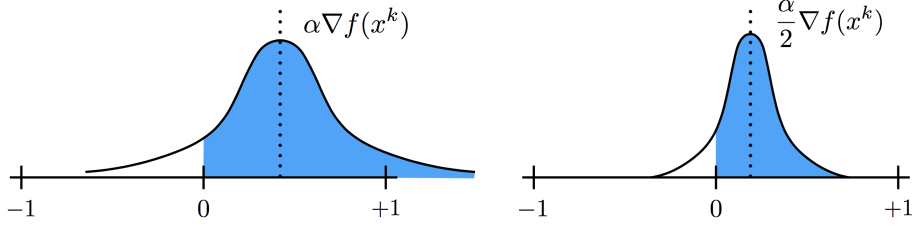

Figure 1: The SR method starts at some location $x$ (in this case 0), adds a perturbation to $x$, and then rounds. As the learning rate $\alpha$ gets smaller, the distribution of the perturbation gets "squished" near the origin, making the algorithm less likely to move. The "squishing" effect is the same for the part of the distribution lying to the left and to the right of $x$, and so it does not effect the *relative* probability of moving left or right.

scale $\Delta$, one would expect BC to perform fundamentally better than SR. While this may seem like a restrictive condition, there is evidence that even non-convex neural networks become well approximated as a quadratic in the later stages of optimization within a neighborhood of a local minimum [21].

Note, our convergence results on BC are for $w_r$ instead of $w_b$, and these measures of convergence are not directly comparable. It is not possible to bound $w_b$ when BC is used, as the values of $w_b$ may not converge in the usual sense (e.g., in the +/-1 binary case $w_r$ might converge to 0, in which case arbitrarily small perturbations to $w_r$ might send $w_b$ to +1 or -1).

## 5 What About Non-Convex Problems?

The global convergence results presented above for convex problems show that, in general, both the SR and BC algorithms converge to within $\mathcal{O}(\Delta)$ accuracy of the minimizer (in expected value). However, these results do not explain the large differences between these methods when applied to non-convex neural nets. We now study how the long-term behavior of SR differs from BC. Note that this section makes no convexity assumptions, and the proposed theoretical results are directly applicable to neural networks.

Typical (continuous-valued) SGD methods have an important exploration-exploitation tradeoff. When the learning rate is large, the algorithm explores by moving quickly between states. Exploitation happens when the learning rate is small. In this case, noise averaging causes the algorithm more greedily pursues local minimizers with lower loss values. Thus, the distribution of iterates produced by the algorithm becomes increasingly concentrated near minimizers as the learning rate vanishes (see, e.g., the large-deviation estimates in [22]). BC maintains this property as well—indeed, we saw in Corollary 1 a class of problems for which the iterates concentrate on the minimizer for small $\alpha_t$.

In this section, we show that the SR method lacks this important tradeoff: as the stepsize gets small and the algorithm slows down, the quality of the iterates produced by the algorithm does *not* improve, and the algorithm does *not* become progressively more likely to produce low-loss iterates. This behavior is illustrated in Figures 1 and 2.

To understand this problem conceptually, consider the simple case of a one-variable optimization problem starting at $x^0 = 0$ with $\Delta = 1$ (Figure 1). On each iteration, the algorithm computes a stochastic approximation $\nabla \tilde{f}$ of the gradient by sampling from a distribution, which we call $p$. This gradient is then multiplied by the stepsize to get $\alpha \nabla \tilde{f}$. The probability of moving to the right (or left) is then roughly proportional to the magnitude of $\alpha \nabla \tilde{f}$. Note the random variable $\alpha \nabla \tilde{f}$ has distribution $p_\alpha(z) = \alpha^{-1} p(z/\alpha)$.

Now, suppose that $\alpha$ is small enough that we can neglect the tails of $p_\alpha(z)$ that lie outside the interval $[-1, 1]$. The probability of transitioning from $x^0 = 0$ to $x^1 = 1$ using stochastic rounding, denoted by $T_\alpha(0, 1)$, is then

$$T_\alpha(0,1) \approx \int_0^1 z p_\alpha(z) dz = \frac{1}{\alpha} \int_0^1 z p(z/\alpha)\, dz = \alpha \int_0^{1/\alpha} p(x) x\, dx \approx \alpha \int_0^\infty p(x) x\, dx,$$

where the first approximation is because we neglected the unlikely case that $\alpha \nabla \tilde{f} > 1$, and the second approximation appears because we added a small tail probability to the estimate. These

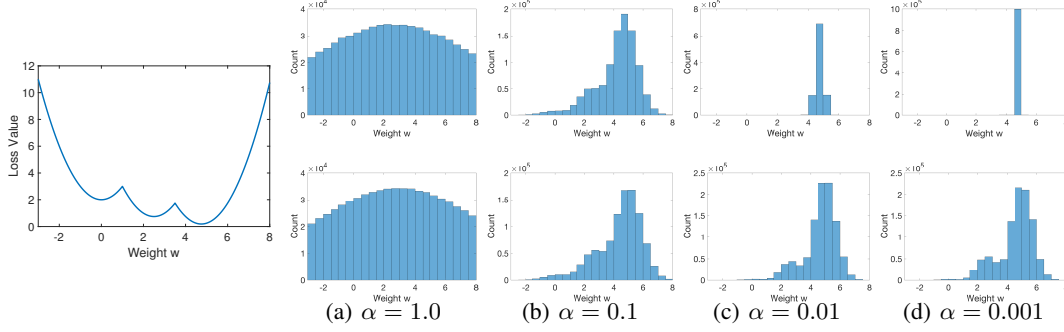

(a) $\alpha = 1.0$  (b) $\alpha = 0.1$  (c) $\alpha = 0.01$  (d) $\alpha = 0.001$

Figure 2: Effect of shrinking the learning rate in SR vs BC on a toy problem. The left figure plots the objective function (8). Histograms plot the distribution of the quantized weights over $10^6$ iterations. The top row of plots correspond to BC, while the bottom row is SR, for different learning rates $\alpha$. As the learning rate $\alpha$ shrinks, the BC distribution concentrates on a minimizer, while the SR distribution stagnates.

approximations get more accurate for small $\alpha$. We see that, assuming the tails of $p$ are "light" enough, we have $T_\alpha(0,1) \sim \alpha \int_0^\infty p(x)x\,dx$ as $\alpha \to 0$. Similarly, $T_\alpha(0,-1) \sim \alpha \int_{-\infty}^0 p(x)x\,dx$ as $\alpha \to 0$.

What does this observation mean for the behavior of SR? First of all, the probability of leaving $x^0$ on an iteration is

$$T_\alpha(0,-1) + T_\alpha(0,1) \approx \alpha \left[ \int_0^\infty p(x)x\,dx + \int_{-\infty}^0 p(x)x\,dx \right],$$

which vanishes for small $\alpha$. This means the algorithm slows down as the learning rate drops off, which is not surprising. However, the *conditional* probability of ending up at $x^1 = 1$ given that the algorithm *did* leave $x^0$ is

$$T_\alpha(0,1|x^1 \neq x^0) \approx \frac{T_\alpha(0,1)}{T_\alpha(0,-1) + T_\alpha(0,1)} = \frac{\int_0^\infty p(x)x\,dx}{\int_{-\infty}^0 p(x)x\,dx + \int_0^\infty p(x)x\,dx},$$

which does not depend on $\alpha$. In other words, provided $\alpha$ is small, SR, on average, makes the same decisions/transitions with learning rate $\alpha$ as it does with learning rate $\alpha/10$; it just takes 10 times longer to make those decisions when $\alpha/10$ is used. In this situation, there is no exploitation benefit in decreasing $\alpha$.

## 5.1 Toy Problem

To gain more intuition about the effect of shrinking the learning rate in SR vs BC, consider the following simple 1-dimensional non-convex problem:

$$\min_w f(w) := \begin{cases} w^2 + 2, & \text{if } w < 1, \\ (w - 2.5)^2 + 0.75, & \text{if } 1 \leq w < 3.5, \\ (w - 4.75)^2 + 0.19, & \text{if } w \geq 3.5. \end{cases} \tag{8}$$

Figure 2 shows a plot of this loss function. To visualize the distribution of iterates, we initialize at $w = 4.0$, and run SR and BC for $10^6$ iterations using a quantization resolution of 0.5.

Figure 2 shows the distribution of the quantized weight parameters $w$ over the iterations when optimized with SR and BC for different learning rates $\alpha$. As we shift from $\alpha = 1$ to $\alpha = 0.001$, the distribution of BC iterates transitions from a wide/explorative distribution to a narrow distribution in which iterates aggressively concentrate on the minimizer. In contrast, the distribution produced by SR concentrates only slightly and then stagnates; the iterates are spread widely even when the learning rate is small.

## 5.2 Asymptotic Analysis of Stochastic Rounding

The above argument is intuitive, but also informal. To make these statements rigorous, we interpret the SR method as a Markov chain. On each iteration, SR starts at some state (iterate) $x$, and moves to

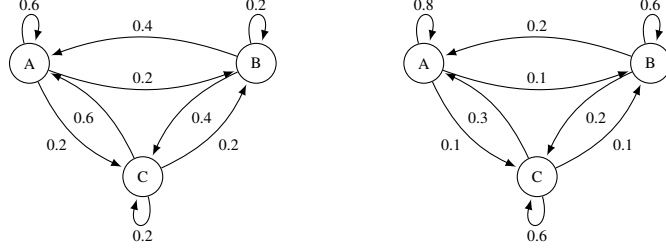

Figure 3: Markov chain example with 3 states. In the right figure, we halved each transition probability for moving between states, with the remaining probability put on the self-loop. Notice that halving all the transition probabilities would not change the equilibrium distribution, and instead would only increase the mixing time of the Markov chain.

a new state $y$ with some transition probability $T_\alpha(x, y)$ that depends only on $x$ and the learning rate $\alpha$. For fixed $\alpha$, this is clearly a Markov process with transition matrix[2] $T_\alpha(x, y)$.

The long-term behavior of this Markov process is determined by the *stationary distribution* of $T_\alpha(x, y)$. We show below that for small $\alpha$, the stationary distribution of $T_\alpha(x, y)$ is nearly invariant to $\alpha$, and thus decreasing $\alpha$ below some threshold has virtually no effect on the long term behavior of the method. This happens because, as $\alpha$ shrinks, the relative transition probabilities remain the same (conditioned on the fact that the parameters change), even though the absolute probabilities decrease (see Figure 3). In this case, there is no exploitation benefit to decreasing $\alpha$.

**Theorem 5.** *Let $p_{x,k}$ denote the probability distribution of the kth entry in $\nabla \tilde{f}(x)$, the stochastic gradient estimate at $x$. Assume there is a constant $C_1$ such that for all $x$, $k$, and $\nu$ we have $\int_\nu^\infty p_{x,k}(z)\, dz \leq \frac{C_1}{\nu^2}$, and some $C_2$ such that both $\int_0^{C_2} p_{x,k}(z)\, dz > 0$ and $\int_{-C_2}^0 p_{x,k}(z)\, dz > 0$. Define the matrix*

$$\tilde{U}(x, y) = \begin{cases} \int_0^\infty p_{x,k}(z)\frac{z}{\Delta}\, dz, \text{if } x \text{ and } y \text{ differ only at coordinate } k, \text{ and } y_k = x_k + \Delta \\ \int_{-\infty}^0 p_{x,k}(z)\frac{z}{\Delta}\, dz, \text{if } x \text{ and } y \text{ differ only at coordinate } k, \text{ and } y_k = x_k - \Delta \\ 0, \text{otherwise,} \end{cases}$$

*and the associated markov chain transition matrix*

$$\tilde{T}_{\alpha_0} = I - \alpha_0 \cdot \mathrm{diag}(\mathbf{1}^T \tilde{U}) + \alpha_0 \tilde{U}, \tag{9}$$

*where $\alpha_0$ is the largest constant that makes $\tilde{T}_{\alpha_0}$ non-negative. Suppose $\tilde{T}_\alpha$ has a stationary distribution, denoted $\tilde{\pi}$. Then, for sufficiently small $\alpha$, $T_\alpha$ has a stationary distribution $\pi_\alpha$, and*

$$\lim_{\alpha \to 0} \pi_\alpha = \tilde{\pi}.$$

*Furthermore, this limiting distribution satisfies $\tilde{\pi}(x) > 0$ for any state $x$, and is thus not concentrated on local minimizers of $f$.*

While the long term stationary behavior of SR is relatively insensitive to $\alpha$, the convergence speed of the algorithm is not. To measure this, we consider the *mixing time* of the Markov chain. Let $\pi_\alpha$ denote the stationary distribution of a Markov chain. We say that the $\epsilon$-mixing time of the chain is $M_\epsilon$ if $M_\epsilon$ is the smallest integer such that [23]

$$|\mathbb{P}(x^{M_\epsilon} \in A | x^0) - \pi(A)| \leq \epsilon, \quad \text{for all } x^0 \text{ and all subsets of states } A \subseteq X. \tag{10}$$

We show below that the mixing time of the Markov chain gets large for small $\alpha$, which means exploration slows down, even though no exploitation gain is being realized.

**Theorem 6.** *Let $p_{x,k}$ satisfy the assumptions of Theorem 5. Choose some $\epsilon$ sufficiently small that there exists a* proper *subset of states $A \subset X$ with stationary probability $\pi_\alpha(A)$ greater than $\epsilon$. Let $M_\epsilon(\alpha)$ denote the $\epsilon$-mixing time of the chain with learning rate $\alpha$. Then,*

$$\lim_{\alpha \to 0} M_\epsilon(\alpha) = \infty.$$

Table 1: Top-1 test error after training with full-precision (ADAM), binarized weights (R-ADAM, SR-ADAM, BC-ADAM), and binarized weights with big batch size (Big SR-ADAM).

| | CIFAR-10 | | | | CIFAR-100 | ImageNet |
|---|---|---|---|---|---|---|
| | VGG-9 | VGG-BC | ResNet-56 | WRN-56-2 | ResNet-56 | ResNet-18 |
| ADAM | 7.97 | 7.12 | 8.10 | 6.62 | 33.98 | 36.04 |
| BC-ADAM | 10.36 | 8.21 | 8.83 | 7.17 | 35.34 | 52.11 |
| Big SR-ADAM | 16.95 | 16.77 | 19.84 | 16.04 | 50.79 | 77.68 |
| SR-ADAM | 23.33 | 20.56 | 26.49 | 21.58 | 58.06 | 88.86 |
| R-ADAM | 23.99 | 21.88 | 33.56 | 27.90 | 68.39 | 91.07 |

## 6 Experiments

To explore the implications of the theory above, we train both VGG-like networks [24] and Residual networks [25] with binarized weights on image classification problems. On CIFAR-10, we train ResNet-56, wide ResNet-56 (WRN-56-2, with 2X more filters than ResNet-56), VGG-9, and the high capacity VGG-BC network used for the original BC model [5]. We also train ResNet-56 on CIFAR-100, and ResNet-18 on ImageNet [26].

We use Adam [27] as our baseline optimizer as we found it to frequently give better results than well-tuned SGD (an observation that is consistent with previous papers on quantized models [1–5]), and we train with the three quantized algorithms mentioned in Section 3, i.e., R-ADAM, SR-ADAM and BC-ADAM. The image pre-processing and data augmentation procedures are the same as [25]. Following [3], we only quantize the weights in the convolutional layers, but not linear layers, during training (See Appendix H.1 for a discussion of this issue, and a detailed description of experiments).

We set the initial learning rate to 0.01 and decrease the learning rate by a factor of 10 at epochs 82 and 122 for CIFAR-10 and CIFAR-100 [25]. For ImageNet experiments, we train the model for 90 epochs and decrease the learning rate at epochs 30 and 60. See Appendix H for additional experiments.

**Results**  The overall results are summarized in Table 1. The binary model trained by BC-ADAM has comparable performance to the full-precision model trained by ADAM. SR-ADAM outperforms R-ADAM, which verifies the effectiveness of Stochastic Rounding. There is a performance gap between SR-ADAM and BC-ADAM across all models and datasets. This is consistent with our theoretical results in Sections 4 and 5, which predict that keeping track of the real-valued weights as in BC-ADAM should produce better minimizers.

**Exploration vs exploitation tradeoffs**  Section 5 discusses the exploration/exploitation tradeoff of continuous-valued SGD methods and predicts that fully discrete methods like SR are unable to enter a greedy phase. To test this effect, we plot the percentage of changed weights (signs different from the initialization) as a function of the training epochs (Figures 4 and 5). SR-ADAM explores aggressively; it changes more weights in the conv layers than both R-ADAM and BC-ADAM, and keeps changing weights until nearly 40% of the weights differ from their starting values (in a binary model, randomly re-assigning weights would result in 50% change). The BC method never changes more than 20% of the weights (Fig 4(b)), indicating that it stays near a local minimizer and explores less. Interestingly, we see that the weights of the conv layers were not changed at all by R-ADAM; when the tails of the stochastic gradient distribution are light, this method is ineffective.

### 6.1  A Way Forward: Big Batch Training

We saw in Section 5 that SR is unable to exploit local minima because, for small learning rates, shrinking the learning rate does not produce additional bias towards moving downhill. This was illustrated in Figure 1. If this is truly the cause of the problem, then our theory predicts that we can improve the performance of SR for low-precision training by increasing the batch size. This shrinks the variance of the gradient distribution in Figure 1 without changing the mean and concentrates more of the gradient distribution towards downhill directions, making the algorithm more greedy.

To verify this, we tried different batch sizes for SR including 128, 256, 512 and 1024, and found that the larger the batch size, the better the performance of SR. Figure 5(a) illustrates the effect of a batch size of 1024 for BC and SR methods. We find that the BC method, like classical SGD, performs best

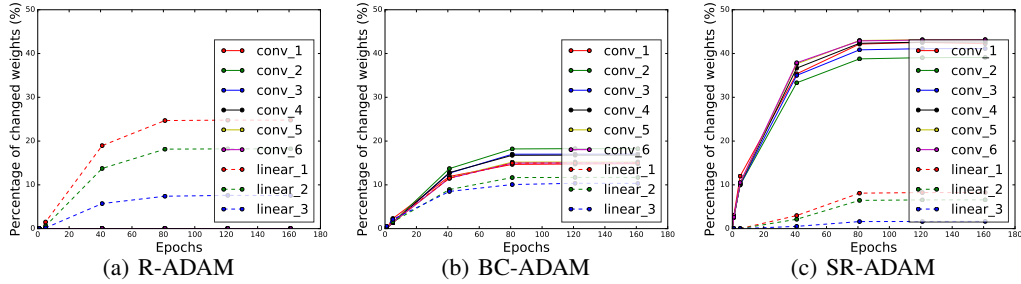

Figure 4: Percentage of weight changes during training of VGG-BC on CIFAR-10.

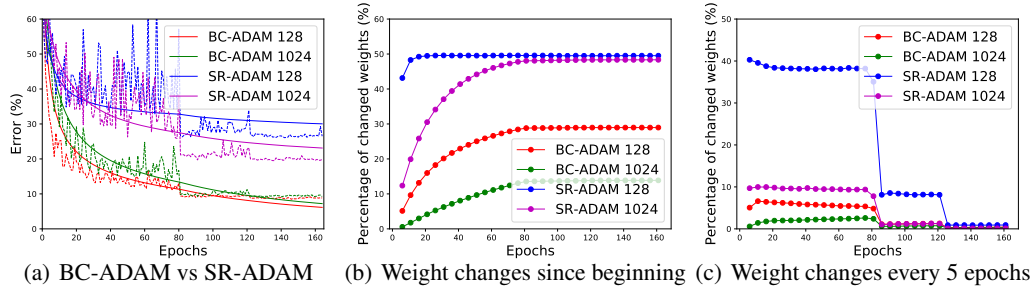

Figure 5: Effect of batch size on SR-ADAM when tested with ResNet-56 on CIFAR-10. (a) Test error vs epoch. Test error is reported with dashed lines, train error with solid lines. (b) Percentage of weight changes since initialization. (c) Percentage of weight changes per every 5 epochs.

with a small batch size. However, a large batch size is essential for the SR method to perform well. Figure 5(b) shows the percentage of weights changed by SR and BC during training. We see that the large batch methods change the weights less aggressively than the small batch methods, indicating less exploration. Figure 5(c) shows the percentage of weights changed during each 5 epochs of training. It is clear that small-batch SR changes weights much more frequently than using a big batch. This property of big batch training clearly benefits SR; we see in Figure 5(a) and Table 1 that big batch training improved performance over SR-ADAM consistently.

In addition to providing a means of improving fixed-point training, this suggests that recently proposed methods using big batches [28, 29] may be able to exploit lower levels of precision to further accelerate training.

## 7 Conclusion

The training of quantized neural networks is essential for deploying machine learning models on portable and ubiquitous devices. We provide a theoretical analysis to better understand the BinaryConnect (BC) and Stochastic Rounding (SR) methods for training quantized networks. We proved convergence results for BC and SR methods that predict an accuracy bound that depends on the coarseness of discretization. For general non-convex problems, we proved that SR differs from conventional stochastic methods in that it is unable to exploit greedy local search. Experiments confirm these findings, and show that the mathematical properties of SR are indeed observable (and very important) in practice.

## Acknowledgments

T. Goldstein was supported in part by the US National Science Foundation (NSF) under grant CCF-1535902, by the US Office of Naval Research under grant N00014-17-1-2078, and by the Sloan Foundation. C. Studer was supported in part by Xilinx, Inc. and by the US NSF under grants ECCS-1408006, CCF-1535897, and CAREER CCF-1652065. H. Samet was supported in part by the US NSF under grant IIS-13-20791.

## Footnotes

*Equal contribution. Author ordering determined by a cryptographically secure random number generator.

[2]Our analysis below does not require the state space to be finite, so $T_\alpha(x, y)$ may be a linear operator rather than a matrix. Nonetheless, we use the term "matrix" as it is standard.

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
