[Supplementary Material · supplementary.pdf]

# Training Quantized Nets: A Deeper Understanding
## Appendices

Here we present proofs of the lemmas and theorems presented in the main paper, as well as some additional experimental details and results.

## A   Proof of Lemma 1

*Proof.* We want to bound the quantization error $r^t$. Consider the $i$-th entry in $r^t$ denoted by $r_i^t$. Similarly, we define $w_i^t$ and $\nabla \tilde{f}(w^t)_i$. Choose some random number $p \in [0,1]$. The stochastic rounding operation produces a value of $r^t$ given by

$$
\begin{aligned}
r_i^t &= Q_s(w_i^t - \alpha_t \nabla \tilde{f}(w^t)_i) - w_i^t + \alpha_t \nabla \tilde{f}(w^t)_i \\
&= \Delta \cdot
\begin{cases}
\dfrac{\alpha_t \nabla \tilde{f}(w^t)_i}{\Delta} + \left\lfloor \dfrac{-\alpha_t \nabla \tilde{f}(w^t)_i}{\Delta} \right\rfloor + 1, & \text{for } p \le -\dfrac{\alpha_t \nabla \tilde{f}(w^t)_i}{\Delta} - \left\lfloor \dfrac{-\alpha_t \nabla \tilde{f}(w^t)_i}{\Delta} \right\rfloor, \\
\dfrac{\alpha_t \nabla \tilde{f}(w^t)_i}{\Delta} + \left\lfloor \dfrac{-\alpha_t \nabla \tilde{f}(w^t)_i}{\Delta} \right\rfloor, & \text{otherwise,}
\end{cases} \\
&= \Delta \cdot
\begin{cases}
-q + 1, & \text{for } p \le q, \\
-q, & \text{otherwise,}
\end{cases}
\end{aligned}
$$

where we write $q = -\dfrac{\alpha_t \nabla \tilde{f}(w^t)_i}{\Delta} - \left\lfloor \dfrac{-\alpha_t \nabla \tilde{f}(w^t)_i}{\Delta} \right\rfloor$ and $q \in [0,1]$.

Now we have

$$
\begin{aligned}
\mathbb{E}_p\left[(r_i^t)^2\right] &\le \Delta^2 ((-q+1)^2 q + (-q)^2 (1-q)) \\
&= \Delta^2 q (1-q) \\
&\le \Delta^2 \min\{q, 1-q\}.
\end{aligned}
$$

Because $\min\{q, 1-q\} \le \left| \dfrac{\alpha_t \nabla \tilde{f}(w^t)_i}{\Delta} \right|$, it follows that $\mathbb{E}_p\left[(r_i^t)^2\right] \le \Delta^2 \left| \dfrac{\alpha_t \nabla \tilde{f}(w^t)_i}{\Delta} \right| \le \Delta \left| \alpha_t \nabla \tilde{f}(w^t)_i \right|$.

Summing over the index $i$ yields

$$
\begin{aligned}
\mathbb{E}_p \big\| r^t \big\|_2^2 &\le \Delta \alpha_t \big\| \nabla \tilde{f}(w^t) \big\|_1 \\
&\le \sqrt{d} \alpha_t \Delta \big\| \nabla \tilde{f}(w^t) \big\|_2 .
\end{aligned} \tag{11}
$$

Now, $\left( \mathbb{E} \big\| \nabla \tilde{f}(w^t) \big\|_2 \right)^2 \le \mathbb{E} \big\| \nabla \tilde{f}(w^t) \big\|_2^2 \le G^2$. Plugging this into (11) yields

$$
\mathbb{E} \big\| r^t \big\|_2^2 \le \sqrt{d} \Delta \alpha_t G. \tag{12}
$$

$\square$

## B   Proof of Theorem 1

*Proof.* From the update rule (6), we get:

$$
\begin{aligned}
w^{t+1} &= Q\big(w^t - \alpha_t \nabla \tilde{f}(w^t)\big) \\
&= w^t - \alpha_t \nabla \tilde{f}(w^t) + r^t,
\end{aligned}
$$

where $r^t$ denotes the quantization used on the $t$-th iteration. Subtracting by the optimal $w^\star$, taking norm, and taking expectation conditioned on $w^t$, we get:

$$
\begin{aligned}
\mathbb{E}\|w^{t+1} - w^\star\|^2 &= \|w^t - w^\star\|^2 - 2\mathbb{E}\langle w^t - w^\star, \alpha_t \nabla \tilde{f}(w^t) - r^t \rangle + \mathbb{E}\|\alpha_t \nabla \tilde{f}(w^t) - r^t\|^2 \\
&= \|w^t - w^\star\|^2 - 2\alpha_t \langle w^t - w^\star, \nabla F(w^t) \rangle + \alpha_t^2 \mathbb{E}\|\nabla \tilde{f}(w^t)\|^2 + \mathbb{E}\|r^t\|^2 \\
&\le \|w^t - w^\star\|^2 - 2\alpha_t \langle w^t - w^\star, \nabla F(w^t) \rangle + \alpha_t^2 G^2 + \sqrt{d}\Delta \alpha_t G,
\end{aligned}
$$

where we use the bounded variance assumption, $\mathbb{E}[r^t] = 0$, and Lemma 1. Using the assumption that $F$ is $\mu$-strongly convex, we can simplify this to:

$$
\mathbb{E}\|w^{t+1} - w^\star\|^2 \le (1 - \alpha_t \mu)\|w^t - w^\star\|^2 - 2\alpha_t (F(w^t) - F(w^\star)) + \alpha_t^2 G^2 + \sqrt{d}\Delta \alpha_t G.
$$

Re-arranging the terms, and taking expectation we get:

$$2\alpha_t \mathbb{E}(F(w^t) - F(w^\star)) \leq (1 - \alpha_t \mu)\mathbb{E}\|w^t - w^\star\|^2 - \mathbb{E}\|w^{t+1} - w^\star\|^2 + \alpha_t^2 G^2 + \sqrt{d}\Delta \alpha_t G.$$

$$\Rightarrow \quad \mathbb{E}(F(w^t) - F(w^\star)) \leq \left(\frac{1}{2\alpha_t} - \frac{\mu}{2}\right)\mathbb{E}\|w^t - w^\star\|^2 - \frac{1}{2\alpha_t}\mathbb{E}\|w^{t+1} - w^\star\|^2 + \frac{\alpha_t}{2}G^2 + \frac{\sqrt{d}\Delta G}{2}.$$

Assume that the stepsize decreases with the rate $\alpha_t = 1/\mu(t+1)$. Then we have:

$$\mathbb{E}(F(w^t) - F(w^\star)) \leq \frac{\mu t}{2}\mathbb{E}\|w^t - w^\star\|^2 - \frac{\mu(t+1)}{2}\mathbb{E}\|w^{t+1} - w^\star\|^2 + \frac{1}{2\mu(t+1)}G^2 + \frac{\sqrt{d}\Delta G}{2}.$$

Averaging over $t = 0$ to $T$, we get a telescoping sum on the right hand side, which yields:

$$\frac{1}{T}\sum_{t=0}^{T}\mathbb{E}(F(w^t) - F(w^\star)) \leq \frac{G^2}{2\mu T}\sum_{t=0}^{T}\frac{1}{t+1} + \frac{\sqrt{d}\Delta G}{2} - \frac{\mu(T+1)}{2}\mathbb{E}\|w^{T+1} - w^\star\|^2$$

$$\leq \frac{(1 + \log(T+1))G^2}{2\mu T} + \frac{\sqrt{d}\Delta G}{2}.$$

Using Jensen's inequality, we have:

$$\mathbb{E}(F(\bar{w}^T) - F(w^\star)) \leq \frac{1}{T}\sum_{t=0}^{T}\mathbb{E}(F(w^t) - F(w^\star)),$$

where $\bar{w}^T = \frac{1}{T}\sum_{t=0}^{T}w^t$, the average of the iterates.

Thus the final convergence theorem is given by:

$$\mathbb{E}[F(\bar{w}^T) - F(w^\star)] \leq \frac{(1 + \log(T+1))G^2}{2\mu T} + \frac{\sqrt{d}\Delta G}{2}.$$

$\square$

## C    Proof of Theorem 2

*Proof.*  From the update rule (6), we have,

$$w^{t+1} = Q\left(w^t - \alpha_t \nabla \tilde{f}(w^t)\right) = w^t - \alpha_t \nabla \tilde{f}(w^t) + r^t,$$

where $r^t$ denotes the quantization error on the $t$-th iteration. Hence we have

$$\|w^{t+1} - w^\star\|^2 = \|w^t - \alpha_t \nabla \tilde{f}(w^t) + r^t - w^\star\|^2$$

$$= \|w^t - w^\star\|^2 - 2\langle w^t - w^\star, \alpha_t \nabla \tilde{f}(w^t) - r^t\rangle + \|\alpha_t \nabla \tilde{f}(w^t) - r^t\|^2.$$

Taking expectation, and using $\mathbb{E}[\tilde{f}(w^t)] = \nabla F(w^t)$ and $\mathbb{E}[r^t] = 0$, we have

$$\mathbb{E}\|w^{t+1} - w^\star\|^2 = \mathbb{E}\|w^t - w^\star\|^2 - 2\alpha_t \mathbb{E}\langle w^t - w^\star, \nabla F(w^t)\rangle + \mathbb{E}\|\alpha_t \nabla \tilde{f}(w^t) - r^t\|^2$$

$$= \mathbb{E}\|w^t - w^\star\|^2 - 2\alpha_t \mathbb{E}\langle w^t - w^\star, \nabla F(w^t)\rangle + \alpha_t^2\mathbb{E}\|\nabla \tilde{f}(w^t)\|^2 + \mathbb{E}\|r^t\|^2.$$

Using the bounded variance assumption $\mathbb{E}\|\nabla \tilde{f}(w^t)\|^2 \leq G^2$ and bounded quantization error in Lemma 1, we have

$$\mathbb{E}\|w^{t+1} - w^\star\|^2 \leq \mathbb{E}\|w^t - w^\star\|^2 - 2\alpha_t \mathbb{E}\langle w^t - w^\star, \nabla F(w^t)\rangle + \alpha_t^2 G^2 + \sqrt{d}\Delta \alpha_t G. \qquad (13)$$

$F(x)$ is convex and hence $\langle \nabla F(x), x_t - x^\star\rangle \geq F(x_t) - F(x^*)$, which can be used in (13) to get

$$\mathbb{E}\|w^{t+1} - w^\star\|^2 \leq \mathbb{E}\|w^t - w^\star\|^2 - 2\alpha_t \mathbb{E}[F(w^t) - F(w^\star)] + \alpha_t^2 G^2 + \sqrt{d}\Delta \alpha_t G.$$

Re-arranging the terms, we have,

$$\mathbb{E}[F(w^t) - F(w^\star)] \leq \frac{1}{2\alpha_t}\left(\mathbb{E}\|w^t - w^\star\|^2 - \mathbb{E}\|w^{t+1} - w^\star\|^2\right) + \frac{\alpha_t}{2}G^2 + \frac{1}{2}\sqrt{d}\Delta G.$$

Accumulate from $t = 1$ to $T$ to get

$$\sum_{t=1}^{T}\mathbb{E}[F(w^t) - F(w^\star)] \leq \frac{1}{2\alpha_1}\mathbb{E}\|w_1 - w^\star\|^2 + \sum_{t=1}^{T}\left(\frac{1}{2\alpha_t} - \frac{1}{2\alpha_{t-1}}\right)\mathbb{E}\|w^t - w^\star\|^2$$

$$+ \sum_{t=1}^{T}\frac{\alpha_t}{2}G^2 + \frac{T}{2}\sqrt{d}\Delta G.$$

Applying $\mathbb{E}\|w^t - w^\star\|^2 \le D^2$ and $\sum_{t=1}^T \alpha_t \le c\sqrt{T+1}$, we have

$$\sum_{t=1}^T \mathbb{E}[F(w^t) - F(w^\star)] \le \frac{\sqrt{T}}{2c}D^2 + \frac{c\sqrt{T+1}}{2}G^2 + \frac{T}{2}\sqrt{d}\Delta G. \tag{14}$$

Since $F(w)$ is convex, we can set $\bar{w}^T = \frac{1}{T}\sum_{t=1}^T w^t$, and use Jensen's inequality to arrive at

$$\mathbb{E}[F(\bar{w}^T) - F(w^\star)] \le \frac{1}{T}\sum_{t=1}^T \mathbb{E}[F(w^t) - F(w^\star)]. \tag{15}$$

Combine (14) and (15) to achieve

$$\mathbb{E}[F(\bar{w}^T) - F(w^\star)] \le \frac{1}{2c\sqrt{T}}D^2 + \frac{\sqrt{T+1}}{2T}cG^2 + \frac{\sqrt{d}\Delta G}{2}.$$

$\square$

# D  Proof of Theorem 3

*Proof.* From the update rule (7), we have,

$$w^{t+1} = w^t - \alpha_t \nabla \tilde{f}\big(Q(w^t)\big) = w^t - \alpha_t \nabla \tilde{f}\big(w^t + r^t\big).$$

Taking expectation conditioned on $w^t$ and $r^t$, we have

$\mathbb{E}\|w^{t+1} - w^\star\|^2$
$= \mathbb{E}\|w^t - \alpha_t \nabla \tilde{f}\big(w^t + r^t\big) - w^\star\|^2$
$= \mathbb{E}\|w^t - \alpha_t \nabla \tilde{f}\big(w^t\big) + \alpha_t \nabla \tilde{f}\big(w^t\big) - \alpha_t \nabla \tilde{f}\big(w^t + r^t\big) - w^\star\|^2$
$= \|w^t - w^\star\|^2 - 2\alpha_t \mathbb{E}\langle w^t - w^\star, \nabla \tilde{f}\big(w^t\big)\rangle + 2\alpha_t \mathbb{E}\langle w^t - w^\star, \nabla \tilde{f}\big(w^t\big) - \nabla \tilde{f}\big(w^t + r^t\big)\rangle + \mathbb{E}\|\alpha_t \nabla \tilde{f}\big(w^t + r^t\big)\|^2$
$= \|w^t - w^\star\|^2 - 2\alpha_t \langle w^t - w^\star, \nabla F\big(w^t\big)\rangle + 2\alpha_t \langle w^t - w^\star, \nabla F\big(w^t\big) - \nabla F\big(w^t + r^t\big)\rangle + \alpha_t^2 \mathbb{E}\|\nabla \tilde{f}\big(w^t + r^t\big)\|^2$
$\le \|w^t - w^\star\|^2 - 2\alpha_t \langle w^t - w^\star, \nabla F\big(w^t\big)\rangle + 2\alpha_t \|w^t - w^\star\|\|\nabla F\big(w^t\big) - \nabla F\big(w^t + r^t\big)\| + \alpha_t^2 G^2$
$\le \|w^t - w^\star\|^2 - 2\alpha_t \langle w^t - w^\star, \nabla F\big(w^t\big)\rangle + 2\alpha_t L\|r^t\|\|w^t - w^\star\| + \alpha_t^2 G^2.$

Using $\|r^t\| \le \sqrt{d}\Delta$ and the bounded domain assumption, we get

$$\mathbb{E}\|w^{t+1} - w^\star\|^2 \le \|w^t - w^\star\|^2 - 2\alpha_t \langle w^t - w^\star, \nabla F\big(w^t\big)\rangle + 2\alpha_t L\sqrt{d}\Delta\|w^t - w^\star\| + \alpha_t^2 G^2$$
$$\le \|w^t - w^\star\|^2 - 2\alpha_t \langle w^t - w^\star, \nabla F\big(w^t\big)\rangle + 2\alpha_t L\sqrt{d}\Delta D + \alpha_t^2 G^2.$$

Taking expectation, and following the same steps as in Theorem 2, we get the convergence result:

$$\mathbb{E}[F(\bar{w}^T) - F(w^\star)] \le \frac{1}{2c\sqrt{T}}D^2 + \frac{\sqrt{T+1}}{2T}cG^2 + \sqrt{d}\Delta LD.$$

$\square$

# E  Proof of Theorem 4

*Proof.* From the update rule (7), we get

$$w^{t+1} = w^t - \alpha_t \nabla \tilde{f}\big(Q(w^t)\big)$$
$$= w^t - \alpha_t \nabla \tilde{f}\big(w^t + r^t\big)$$
$$= w^t - \alpha_t [\nabla \tilde{f}\big(w^t\big) + \nabla^2 \tilde{f}\big(w^t\big)r^t + \hat{r}^t]$$

where $\|\hat{r}^t\| \le \frac{L_2}{2}\|r^t\|^2$ from our assumption on the Hessian. Note that in general $r^t$ has mean zero while $\hat{r}^t$ does not. Using the same steps as in the Theorem 1, we get

$$\mathbb{E}\|w^{t+1} - w^\star\|^2 = \|w^t - w^\star\|^2 - 2\alpha_t \mathbb{E}\langle w^t - w^\star, \nabla \tilde{f}(w^t + r^t)\rangle + \alpha_t^2 \mathbb{E}\|\nabla \tilde{f}(w^t + r^t)\|^2.$$
$$\le \|w^t - w^\star\|^2 - 2\alpha_t \mathbb{E}\langle w^t - w^\star, \nabla F(w^t) + \hat{r}^t\rangle + \alpha_t^2 G^2$$
$$= \|w^t - w^\star\|^2 - 2\alpha_t \mathbb{E}\langle w^t - w^\star, \nabla F(w^t)\rangle + \alpha_t^2 G^2 - 2\alpha_t \mathbb{E}\langle w^t - w^\star, \hat{r}^t\rangle$$

Assuming the domain has finite diameter $D$, and observing that the quantization error for BC-SGD can always be upper-bounded as $\|r^t\| \leq \sqrt{d}\Delta$, we get:

$$-2\alpha_t \mathbb{E}\langle w^t - w^\star, \hat{r}^t\rangle \leq 2\alpha_t D\mathbb{E}\|\hat{r}^t\| \leq 2\alpha_t D\frac{L_2}{2}\|r^t\| \leq \alpha_t DL_2\sqrt{d}\Delta.$$

Following the same steps as in Theorem 1, we get

$$\mathbb{E}[F(\bar{w}^T) - F(w^\star)] \leq \frac{(1 + \log(T+1))G^2}{2\mu T} + \frac{DL_2\sqrt{d}\Delta}{2}.$$

$\square$

## F   Proof of Theorem 5

*Proof.* Let the matrix $U_\alpha$ be a partial transition matrix defined by $U_\alpha(x,x) = 0$, and $U_\alpha(x,y) = T_\alpha(x,y)$ for $x \neq y$. From $U_\alpha$, we can get back the full transition matrix $T_\alpha$ using the formula

$$T_\alpha = I - \text{diag}(\mathbf{1}^T U_\alpha) + U_\alpha.$$

Note that this formula is essentially "filling in" the diagonal entries of $T_\alpha$ so that every column sums to 1, thus making $T_\alpha$ a valid stochastic matrix.

Let's bound the entries in $U_\alpha$. Suppose that we begin an iteration of the stochastic rounding algorithm at some point $x$. Consider an adjacent point $y$ that differs from $x$ at only 1 coordinate, $k$, with $y_k = x_k + \Delta$. Then we have

$$
\begin{aligned}
U_\alpha(x,y) &= \frac{1}{\alpha}\int_0^\Delta p_{x,k}(x/\alpha)\frac{x}{\Delta}\,dx + \frac{1}{\alpha}\int_\Delta^{2\Delta} p_{x,k}(x/\alpha)\frac{2\Delta - x}{\Delta}\,dx \\
&= \frac{1}{\alpha}\int_0^{\Delta/\alpha} p_{x,k}(z)\frac{\alpha z}{\Delta}\alpha\,dz + \frac{1}{\alpha}\int_{\Delta/\alpha}^{2\Delta/\alpha} p_{x,k}(z)\frac{2\Delta - \alpha z}{\Delta}\alpha\,dz \\
&\leq \alpha\int_0^{\Delta/\alpha} p_{x,k}(z)\frac{z}{\Delta}\,dz + \int_{\Delta/\alpha}^\infty p_{x,k}(z)\,dz \\
&= \alpha\int_0^\infty p_{x,k}(z)\frac{z}{\Delta}\,dz + O(\alpha^2). 
\end{aligned}
\tag{16}
$$

Note we have used the decay assumption:

$$\int_\nu^\infty p_{x,k}(z) \leq \frac{C}{\nu^2}.$$

Likewise, if $y_k = x_k - \Delta$, then the transition probability is

$$U_\alpha(x,y) = \alpha\int_{-\infty}^0 p_{x,k}(z)\frac{z}{\Delta}\,dz + O(\alpha^2), \tag{17}$$

and if $y_k = x_k \pm m\Delta$ for an integer $m > 1$,

$$U_\alpha(x,y) = O(\alpha^2). \tag{18}$$

We can approximate the behavior of $U_\alpha$ using the matrix

$$\tilde{U}(x,y) = \begin{cases} \int_0^\infty p_{x,k}(z)\frac{z}{\Delta}\,dz, & \text{if } x \text{ and } y \text{ differ only at coordinate } k, \text{ and } y_k = x_k + \Delta \\ \int_{-\infty}^0 p_{x,k}(z)\frac{z}{\Delta}\,dz, & \text{if } x \text{ and } y \text{ differ only at coordinate } k, \text{ and } y_k = x_k - \Delta \\ 0, & \text{otherwise.} \end{cases}$$

Define the associated markov chain transition matrix

$$\tilde{T}_{\alpha_0} = I - \alpha_0 \cdot \text{diag}(\mathbf{1}^T\tilde{U}) + \alpha_0\tilde{U}, \tag{19}$$

where $\alpha_0$ is the largest scalar such that the stochastic linear operator $\tilde{T}_{\alpha_0}$ has non-negative entries. For $\alpha < \alpha_0$, $\tilde{T}_\alpha$ has non-negative entries and column sums equal to 1; it thus defines the transition operator of a markov chain. Let $\tilde{\pi}$ denote the stationary distribution of the markov chain with transition matrix $\tilde{T}_{\alpha_0}$.

We now claim that $\tilde{\pi}$ is also the stationary distribution of $\tilde{T}_\alpha$ for all $\alpha < \alpha_0$. We verify this by noting that

$$
\begin{aligned}
\tilde{T}_\alpha &= (I - \alpha \cdot \text{diag}(\mathbf{1}^T\tilde{U})) + \alpha\tilde{U} \\
&= (1 - \frac{\alpha}{\alpha_0})I + \frac{\alpha}{\alpha_0}[I - \alpha_0 \cdot \text{diag}(\mathbf{1}^T\tilde{U}) + \alpha_0\tilde{U}] \\
&= (1 - \frac{\alpha}{\alpha_0})I + \frac{\alpha}{\alpha_0}\tilde{T}_{\alpha_0},
\end{aligned}
\tag{20}
$$

and so $\tilde{T}_\alpha \tilde{\pi} = (1 - \frac{\alpha}{\alpha_0})\tilde{\pi} + \frac{\alpha}{\alpha_0}\tilde{\pi} = \tilde{\pi}$.

Recall that $T_\alpha$ is the transition matrix for the Markov chain generated by the stochastic rounding algorithm with learning rate $\alpha$. We wish to show that this markov chain is well approximated by $\tilde{T}_\alpha$. Note that

$$T_\alpha(x, y) = \prod_{k, x_k \neq y_k} T_\alpha(x, x + (y_k - x_k)\Delta e_k) \leq O(\alpha^2)$$

when $x, y$ differ at more than 1 coordinate. In other words, transitions between multiple coordinates simultaneously become vanishingly unlikely for small $\alpha$. When $x$ and $y$ differ by exactly 1 coordinate, we know from (16) that

$$T_\alpha(x, y) = \alpha U(x, y) + O(\alpha^2).$$

These observations show that the off-diagonal elements of $T_\alpha$ are well approximated (up to uniform $O(\alpha^2)$ error) by the corresponding elements in $\alpha U$. Since the columns of $T_\alpha$ sum to one, the diagonal elements are well approximated as well, and we have

$$T_\alpha = (I - \alpha \cdot \text{diag}(\mathbf{1}^T U)) + \alpha U + O(\alpha^2) = \tilde{T}_\alpha + O(\alpha^2).$$

To be precise, the notation above means that

$$|T_\alpha(x, y) - \tilde{T}_\alpha(x, y)| < C\alpha^2, \tag{21}$$

for some $C$ that is uniform over $(x, y)$.

We are now ready to show that the stationary distribution of $T_\alpha$ exists and approaches $\tilde{\pi}$. Re-arranging (20) gives us

$$\alpha_0 \tilde{T}_\alpha + (\alpha - \alpha_0)I = \alpha \tilde{T}_{\alpha_0}.$$

Combining this with (21), we get

$$\left\| \alpha_0 T_\alpha + (\alpha - \alpha_0)I - \alpha \tilde{T}_{\alpha_0} \right\|_\infty < O(\alpha^2),$$

and so

$$\left\| \frac{\alpha_0}{\alpha} T_\alpha + (1 - \frac{\alpha_0}{\alpha})I - \tilde{T}_{\alpha_0} \right\|_\infty < O(\alpha). \tag{22}$$

From (22), we see that the matrix $\frac{\alpha_0}{\alpha} T_\alpha + (1 - \frac{\alpha_0}{\alpha})I$ approaches $\tilde{T}_{\alpha_0}$. Note that $\tilde{\pi}$ is the Perron-Frobenius eigenvalue of $\tilde{T}_{\alpha_0}$, and thus has multiplicity 1. Multiplicity 1 eigenvalues/vectors of a matrix vary continuously with small perturbations to that matrix (Theorem 8, p130 of [30]). It follows that, for small $\alpha$, $\frac{\alpha_0}{\alpha} T_\alpha + (1 - \frac{\alpha_0}{\alpha})I$ has a stationary distribution, and this distribution approaches $\tilde{\pi}$. The leading eigenvector of $\frac{\alpha_0}{\alpha} T_\alpha + (1 - \frac{\alpha_0}{\alpha})I$ is the same as the leading eigenvector of $T_\alpha$, and it follows that $T_\alpha$ has a stationary distribution that approaches $\tilde{\pi}$.

Finally, note that we have assumed $\int_0^{C_2} p_{x,k}(z)\,dz > 0$ and $\int_{-C_2}^0 p_{x,k}(z)\,dz > 0$. Under this assumption, for $\alpha < \frac{1}{C_2}$, $\tilde{T}_{\alpha_0}(x, y) > 0$ whenever $x, y$ are neighbors the differ at a single coordinate. It follows that every state in the Markov chain $\tilde{T}_{\alpha_0}$ is accessible from every other state by traversing a path of non-zero transition probabilities, and so $\tilde{\pi}(x) > 0$ for every state $x$.

$\square$

# G   Proof of Theorem 6

*Proof.* Given some distribution $\pi$ over the states of the markov chain, and some set $A$ of states, let $[\pi]_A = \sum_{a \in A} \pi(a)$ denote the measure of $A$ with respect to $\pi$.

Suppose for contradiction that the mixing time of the chain remains bounded as $\alpha$ vanishes. Then we can find an integer $M_\epsilon$ that upper bounds the $\epsilon$-mixing time for all $\alpha$. By the assumption of the theorem, we can select some set of states $A$ with $[\tilde{\pi}]_A > \epsilon$, and some starting state $y \notin A$. Let $e$ be a distribution (a vector in the finite-state case) with $e_y = 1$, $e_k = 0$ for $k \neq y$. Note that $[e]_A = 0$ because $y \notin A$. Then

$$\left| [e]_A - [\tilde{\pi}]_A \right| > \epsilon.$$

Note that, as $\alpha \to 0$, we have $\|T_\alpha - \tilde{T}_\alpha\| \to 0$ and thus $\|T_\alpha^{M_\epsilon} - \tilde{T}_\alpha^{M_\epsilon}\| \to 0$. We also see from the definition of $\tilde{T}_\alpha$ in (19), $\lim_{\alpha \to 0} \tilde{T}_\alpha = I$. It follows that

$$\lim_{\alpha \to 0} \left| [T_\alpha^{M_\epsilon} e]_A - [\tilde{\pi}]_A \right| = \left| [e]_A - [\tilde{\pi}]_A \right| > \epsilon,$$

and so for some $\alpha$ the inequality (10) is violated. This is a contradiction because it was assumed $M_\epsilon$ is an upper bound on the mixing time.

$\square$

# H  Additional Experimental Details & Results

## H.1  Neural Net Architecture & Training Details

We train image classifiers using two types of networks, VGG-like CNNs [24] and Residual networks [25], on CIFAR-10/100 [31] and ImageNet 2012 [26]. VGG-9 on CIFAR-10 consists of 7 convolutional layers and 2 fully connected layers. The convolutional layers contain 64, 64, 128, 128, 256, 256 and 256 of $3 \times 3$ filters respectively. There is a Batch Normalization and ReLU after each convolutional layer and the first fully connected layer. The details of the architecture are presented in Table 2. VGG-BC is a high-capacity network used for the original BC method [5], which contains 6 convolutional layers and 3 linear layers. We use the same architecture as in [5] except using softmax and cross-entropy loss instead of SVM and squared hinge loss, respectively. The details of the architecture are presented in Table 3. ResNets-56 has 55 convolutional layers and one linear layer, and contains three stages of residual blocks where each stage has the same number of residual blocks. We also create a wide ResNet-56 (WRN-56-2) that doubles the number of filters in each residual block as in [32]. ResNets-18 for ImageNet has the same description as in [25].

The default minibatch size is 128. However, the big-batch SR-ADAM method adopts a large minibatch size (512 for WRN-56-2 and ResNet-18 and 1024 for other models). Following [5], we do not use weight decay during training. We implement all models in Torch7 [33] and train the quantized models with NVIDIA GPUs.

Similar to [3], we only quantize the weights in the convolutional layers, but not linear layers, during training. Binarizing linear layers causes some performance drop without much computational speedup. This is because fully connected layers have very little computation overhead compared to Conv layers. Also, for state-of-the-art CNNs, the number of FC parameters is quite small. The number of params of Conv/FC layers for CNNs in Table 1 are (in millions): VGG-9: 1.7/1.1, VGG-BC: 4.6/9.4, ResNet-56: 0.84/0.0006, WRN-56-2: 3.4/0.001, ResNet-18: 11.2/0.5. While the VGG-like nets have many FC parameters, the more efficient and higher performing ResNets are almost entirely convolutional.

Table 2: VGG-9 on CIFAR-10.

| layer type | kernel size | input size | output size |
|---|---|---|---|
| Conv_1 | $3 \times 3$ | $3 \times 32 \times 32$ | $64 \times 32 \times 32$ |
| Conv_2 | $3 \times 3$ | $64 \times 32 \times 32$ | $64 \times 32 \times 32$ |
| Max Pooling | $2 \times 2$ | $64 \times 32 \times 32$ | $64 \times 16 \times 16$ |
| Conv_3 | $3 \times 3$ | $64 \times 16 \times 16$ | $128 \times 16 \times 16$ |
| Conv_4 | $3 \times 3$ | $128 \times 16 \times 16$ | $128 \times 16 \times 16$ |
| Max Pooling | $2 \times 2$ | $128 \times 16 \times 16$ | $128 \times 8 \times 8$ |
| Conv_5 | $3 \times 3$ | $128 \times 8 \times 8$ | $256 \times 8 \times 8$ |
| Conv_6 | $3 \times 3$ | $256 \times 8 \times 8$ | $256 \times 8 \times 8$ |
| Conv_7 | $3 \times 3$ | $256 \times 8 \times 8$ | $256 \times 8 \times 8$ |
| Max Pooling | $2 \times 2$ | $256 \times 8 \times 8$ | $256 \times 4 \times 4$ |
| Linear | $1 \times 1$ | $1 \times 4096$ | $1 \times 256$ |
| Linear | $1 \times 1$ | $1 \times 256$ | $1 \times 10$ |

Table 3: VGG-BC for CIFAR-10.

| layer type | kernel size | input size | output size |
|---|---|---|---|
| Conv_1 | $3 \times 3$ | $3 \times 32 \times 32$ | $128 \times 32 \times 32$ |
| Conv_2 | $3 \times 3$ | $128 \times 32 \times 32$ | $128 \times 32 \times 32$ |
| Max Pooling | $2 \times 2$ | $128 \times 32 \times 32$ | $128 \times 16 \times 16$ |
| Conv_3 | $3 \times 3$ | $128 \times 16 \times 16$ | $256 \times 16 \times 16$ |
| Conv_4 | $3 \times 3$ | $256 \times 16 \times 16$ | $256 \times 16 \times 16$ |
| Max Pooling | $2 \times 2$ | $256 \times 16 \times 16$ | $256 \times 8 \times 8$ |
| Conv_5 | $3 \times 3$ | $256 \times 8 \times 8$ | $512 \times 8 \times 8$ |
| Conv_6 | $3 \times 3$ | $512 \times 8 \times 8$ | $512 \times 8 \times 8$ |
| Max Pooling | $2 \times 2$ | $512 \times 8 \times 8$ | $512 \times 4 \times 4$ |
| Linear | $1 \times 1$ | $1 \times 8192$ | $1 \times 1024$ |
| Linear | $1 \times 1$ | $1 \times 1024$ | $1 \times 1024$ |
| Linear | $1 \times 1$ | $1 \times 1024$ | $1 \times 10$ |

## H.2  Convergence Curves

The convergence curves for training and testing errors reported in Table 1 are shown in Figure 6.

Figure 6: Training and testing errors of different training methods for VGG-9, VGG-BC, ResNet-56, Wide-ResNet-56-2 and ResNet-18. The solid line is the training error and the dashed line is the testing error.

## H.3 Weight Initialization and Learning Rate

For experiments on SR-Adam and R-Adam, the weights of convolutional layers are intitialized with random Rademacher (±1) variables. The authors of BC [5] adopt a small initial learning rate (0.003) and it takes 500 epochs to converge. It is observed that large binary weights ($\Delta = 1$) will generate small gradients when batch normalization is used [34], hence a large learning rate is necessary for faster convergence. We experiment with a larger learning rate (0.01) and find it converges to the same performance within 160 epochs, comparing with 500 epochs in the original paper [5].

## H.4 Weight Decay

Figure 7 shows the effect of applying weight decay to BC-ADAM. As shown in Figure 7(a), BC-ADAM with 1e-5 weight decay yields worse performance compared to zero weight decay. Applying weight decay in BC-ADAM will shrink $w_r$ to 0, as well as increase the distance between $w_b$ and $w_r$. Figure 7(b) and 7(c) shows the distance between $w_b$ and $w_r$ during training. With 1e-5 weight decay, the average weight difference between $w_b$ and $w_r$ approaches 1, which indicates $w_r$ is close to zero. Weight decay cannot "decay" the weight of SR as $\|w_b\|_2$ is the same for all binarized networks.

Figure 7: The effect of weight decay (WD) on BC-ADAM for training VGG-BC. The y-axis of (b) and (c) is the averaged weight difference between the binary weights $w_b$ and the real-valued weights $w_r$ , i.e., $\frac{1}{d}\|w_b^t - w_r^t\|_1$. where $d$ is the number of weights in $w_b$.