[Reviews · NeurIPS 2017]

Reviewer 1



This papers investigates theoretically and numerically why the recent BinaryConnect (BC) works better in comparison to more traditional rounding schemes, such as Stochastic Rounding (SR). It proves that for convex functions, (the continuous weights in) BC can converge to the global minimum, while SR methods fair less well. Also, it is proven that, below a certain value, learning in SR is unaffected by decreasing the learning rate, except that the learning process is slowed down. The paper is, to the best of my understanding: 1) Clear, modulo the issues below. 2) Technically correct, except some typos. 3) Deals with a significant topic, which has practical implications: understanding how to train neural in networks with low precision. 4) Novel and original, especially considering that most papers on this subject do not contain much theory. 5) Has interesting results. Specifically, I think it helps clarify why is it so hard to train with SR over BC (it would be extremely useful if one could use SR, since then there would be any need to store the full precision weights during training). Some issues: 1) It is confusing that w_r and w_b are both denoted w in section 4. For example, since the BC bounds are on w_r, it should be clarified that the F(w_b) behaves differently (e.g., it should have an "accuracy floor), and what are the implications (e.g., it seems a somewhat unfair to compare this bound with the SR bound on w_b). 2) It is not clear how tight are these bounds, and especially the accuracy floor. The paper would have been stronger if you it had a lower bound on the error in SR. Also, I would suggest doing a (simple) simulation of a convex/strongly-convex functions to check the tightness of these results. 3) Percentage of weight change graphs do not look very convincing. In figure 3 the linear layers actually decrease in SR in comparison to BC. Also, in figure 4(b) both batch sizes arrive to almost the same value: the different convergence speed could be related to the fact that with smaller batch sizes we do more iterations per epoch. Minor issues: * The separation of the references to various advantages in lines [26-27] seems wrong. For example, [1] actually accelerated inference throughput (using a xnor kenrel), while [3,4] only discussed this. * line 34: "on" -> "to" * It should be mentioned on the description of the methods (R SR BC) that the weights are typically restricted to a finite domain. * lines 100-101: I guess the authors refer to the binary weights here, not w_r, since w_r was restricted to [-1,1] not {-1,1}. * lines 102-103: not clear if this true. See follow-up of [1]: "Quantized Neural Networks: Training Neural Networks with Low Precision Weights and Activations". * Lemma 1: I don't think it was previously defined that d is the dimension of w. Also, in the proof, the authors should consider keeping the L1 norm on the gradients instead of \sqrt(d) times the L2 norm, which can give a much higher "accuracy floor". * Line 122: confusing sentence: When alpha -> 0 the rounding actually becomes more destructive, as shown in section 5. The authors should consider replacing "rounding effect" with "rounding error per step", or removing sentence. * First equation in Supplementary Material (SM): change +1 to -1 * line 334 in SM: (11) -> (10) %% After author feedback %% The authors have addressed my concerns.

Reviewer 2



his paper presents theoretical analysis for understanding the quantized neural networks. Particularly, the convergence analysis is performed for two types of quantizers, including the stochastic rounding (SR) and binary connect (BC), with different assumptions. Empirical evaluation and results are also provided to justify the theoretical results. Pros. 1. In general this paper is well written. Quantized nets have lots of potential applications, especially for the low-power embedded devices. Although many quantized nets have shown promising performance in practice, a rigorous analysis of these models is very essential. This paper presents some interesting theoretical results along this direction. 2. This paper shows the convergence results in both convex and non-convex settings, although certain assumptions are imposed. Cons. 1. Experimental results from Table 1 suggest that BC-ADAM outperforms SR-ADAM and R-ADAM in every case. An additional comparison of runtime behavior would be very helpful in evaluating the efficiency of these methods in practice. 2. The authors claimed that, a smaller batch size leads to an lower error for BC-ADAM, while a larger batch size is preferred for SR-ADAM. However, only two batch sizes (i.e., 128 and 2014) are employed in the evaluation. More batch sizes such as 256 and 512 should be adopted to testify if the conclusion consistently holds across multiple sizes.

Reviewer 3



Summary --------- This paper analyzes the performance of quantized networks at a theoretical level. It proofs convergence results for two types of quantization (stochastic rounding and binary connect) in a convex setting and provides theoretical insights into the superiority of binary connect over stochastic rounding in non-convex applications. General comments —----------------- The paper states that the motivation for quantization is training of neural networks on embedded devices with limited power, memory and/or support of floating-point arithmetic. I do not agree with this argumentation, because we can easily train a neural network on a powerful GPU grid and still do the inexpensive forward-pass on a mobile device. Furthermore, the vast majority of devices does support floating-point arithmetic. And finally, binary connect, which the paper finds to be superior to stochastic rounding, still needs floating-point arithmetic. Presentation ------------- The paper is very well written and structured. The discussion of related work is comprehensive. The notation is clean and consistent. All proofs can be found in the (extensive) supplemental material, which is nice because the reader can appreciate the main results without getting lost in too much detail. Technical section ------------------ The paper’s theoretical contributions can be divided into convex and non-convex results: For convex problems, the paper provides ergodic convergence results for quantization through stochastic rounding (SR) and binary connect (BC). In particular, it shows that both methods behave fundamentally different for strongly convex problems. While I appreciate the importance of those results at a theoretical level, they are limited in their applicability because today’s optimization problems, and neural networks in particular, are rarely convex. For non-convex problems, the paper proofs that SR does not enter an exploitation phase for small learning rates (Theorem 5) but just gets slow (Theorem 6), because its behavior is essentially independent of the learning rate below a certain threshold. The paper claims that this is in contrast to BC, although no formal proof supporting this claim is provided for the non-convex case. Experiments ------------- The experimental evaluation compares the effect of different types of quantization on the performance of a computer vision task. The main purpose of the experiments is to understand the impact of the theoretical results in practice. - First of all, the paper should clearly state the task (image classification) and quantization step (binary). Experimental results for more challenging tasks and finer quantization steps would have been interesting. - Standard SGD with learning rates following a Robbins-Monro sequence should have been used instead of Adam. After all, the theoretical results were not derived for Adam’s adaptive learning rate, which I am concerned might conceal some of the effects of standard SGD. - Table 1 feels a little incoherent, because different network architectures were used for different datasets. - The test error on ImageNet is very high. Is this top-1 accuracy? - Why only quantize convolutional weights (l.222)? The vast majority of weights are in the fully-connected layers, so it is not surprising that the quantized performance is close to the unquantized networks if most weights are the same. - A second experiment computes the percentage of changed weights to give insights into the exploration-exploitation behaviour of SR and BC. I am not sure this is a meaningful experiment. Due to the redundancy in neural networks, changing a large numbers of weights can still result in a very similar function. Also, it seems that the curves in Fig. 3(c) do saturate, meaning that only very few weights are changed between epochs 80 and 100 and indicating exploitation even with SR. Minor comments ----------------- l.76: Citation missing. Theorem 1: w^* not defined (probably global minimizer). l.137: F_i not defined. l.242: The percentage of changed weights is around 40%, not 50%. Conclusion ----------- The paper makes important contributions to the theoretical understanding of quantized nets. Although I am not convinced by the experiments and concerned about the practical relevance, this paper should be accepted due to its strong theoretical contributions.